

# Building Asset Value Mapping in Support of Flood Risk Assessment: A Case Study of Shanghai, China

**Jidong Wu [1], Xu Wang [1], Elco Koks [2]**

[1] State Key Laboratory of Earth Surface Processes and Resource Ecology; Key Laboratory of Environmental Change and Natural Disaster, MOE, Beijing Normal University, Beijing 100875, China
[2] Department of Water & Climate Risk, Institute for Environmental Studies, Vrije Universiteit Amsterdam, De Boelelaan 1085, 1081 HV Amsterdam, The Netherlands

*Correspondence to*: Wu J. (wujidong@bnu.edu.cn)

**Abstract**: Exposure is an integral part of any natural disaster risk assessment. As one of the consequences of natural disasters, damage to buildings is one of the most important concerns. As such, estimates of the building stock and the values at risk can assist natural disaster risk management, including determining the damage extent and severity. Unfortunately, only little information about building asset value is readily available in most countries (especially its spatial distributions) including in China, given that the statistical data on building floor area (BFA) is collected by administrative unities in China. In order to bridge the gap between aggregated census statistical buildings floor-area data to geo-coded building asset value data, this article introduces a methodology for a city-scale building asset value mapping using Shanghai as an example. It consists of a census BFA disaggregation (downscaling) by means of a building footprint map extracted from high-resolution remote sensing data and LandScan population density data, and a financial appraisal of building asset values. A validation with statistical data confirms the feasibility of the modelled building storey. The example of the use of the developed building asset value map in exposure assessment of a flood scenario of Shanghai demonstrated that the dataset offers immense analytical flexibility for flood risk assessment. The method used in this paper is transferable to be applied in other cities of China for building asset value mapping.

## 1 Introduction

Disaster risk is often defined as a function of hazard (i.e. probability of occurrence), exposure (elements at risk) and vulnerability (the susceptibility of those elements) (UNISDR, 2013; De Bono and Mora, 2014). For natural disaster risk analysis, there is still quite often a spatial mismatch between hazard intensity data (e.g., flood depth), frequently modelled on a high-resolution raster level, and exposure data, which is usually only available at coarse census units (e.g. counties) or aggregated land-use/land cover classes (Chen et al., 2004; Thieken et al., 2006; Figueiredo and Martina, 2016). Because buildings and population are tending to be concentrated in the flatter regions and along roads and rivers (Gallup et al., 1999; Felkner and Townsend, 2011), a uniform distribution of exposure data within a county may be pragmatic, but not realistic. As such, the quality of the exposure data, including the building asset value





distribution, is one of the most important uncertainties in flood risk assessments (Merz et al., 2004; Jongman et al., 2012; de Moel and Aerts, 2011, de Moel et al., 2014; Meyer et al., 2013). The main cause for this uncertainty is due to the high spatial heterogeneity of water inundation depth. This implicates that flood risk assessments are much more sensitive to the exposure data resolution than, for instance, earthquake risk assessments (Chen et al., 2004; Thieken et al., 2006). Therefore, developing an exposure model and providing a building asset value map is essential for reducing the uncertainty of the results in the risk model (Gunasekera et al., 2015).

Ideally, acquiring actual building information (e.g., location, footprint, height, structure type, age, etc.) would satisfy the demand of exposure data input for risk analysis. Unfortunately, the availability of such detailed data is still limited. In order to obtain best estimates, several studies have attempted to seek techniques for producing building asset maps. For these estimations, the population distribution is a widely-used proxy variable to disaggregate building asset value from census area to a finer resolution. Thieken et al. (2006), for instance, assume that the population distribution directly reflects the distribution of residential asset values. They found a direct correlation between the two and they therefore use population distribution data and land-cover data as an ancillary variable. This resulted in dasymetric maps that show a unit value of residential assets for the whole Germany for natural disaster risk assessment. Silva et al. (2014) use LandScan population distribution to disaggregate the building stock at parish level for an earthquake risk assessment in mainland Portugal. Finally, Seifert et al. (2010) use LULC data and building density fraction information to disaggregate assets into different economic sectors.

For China, statistical BFA data is available at county level in the population census tabulation. This is the most consistent and reliable source of information about buildings. Moreover, the availability of the actual building information is increasing. As each province of China is building the Map World platform under the instruction of National Administration of Surveying, Mapping and Geoinformation, and provide the electronic map with building footprints generated from high-resolution remote sensing data. Both of the census statistical BFA data and the building footprint data made it possible to produce a geo-coded building asset value map for risk analysis.

For resolving the gap that building stock data often only exists on an administrative scale, which is inconsistent with the distribution of the real hazard extent. This paper aims at developing a methodology to map the building asset value with spatial-resolution at 2.5m×2.5m using Shanghai as a case study. To test the reliability of the disaggregated BFA map, validation is performed with real statistical data. Furthermore, the building asset value map is applied in a flood damage estimation under a flood scenario of Shanghai to testify the flexibility of the BFA map. The remaining paper is structured as follows. Section 2 presents the materials and methods that are used to obtain the building asset value map.



Section 3 presents the results and Section 4 for a discussion about the application of building asset value map in flood loss modelling and uncertainties in this study. Finally, Section 5 presents the conclusion.

## 2 Materials and Methods

### 2.1. Study area

5  Shanghai situates on the eastern fringe of the Yangtze River Delta in the East of China (Figure 1). Due to its unique geographical location, Shanghai is considered to be the most vulnerable city to coastal flooding among the nine deltaic coastal cities worldwide (Balica et al., 2012). Shanghai is a low-lying coastal city with significant subsidence currently at a maximum rate of 24.12 mm/a, as reported by Wang et al. (2012), and with the most developed economy and highest population density in China, Gross Domestic Product exceeded 2.5 trillion China Yuan and total population

10  surpassed 24 million with a population density of 3809 persons per km$^2$ in 2015 for Shanghai. Shanghai's flood risk is receiving increased media coverage and policy attention (Yu et al., 2012; Yin et al., 2013; Hallegatte et al., 2013; Scholten, 2016). A reliable and consistent map with building assets value distribution for Shanghai is much needed for reducing the uncertainty of flood risk assessment.

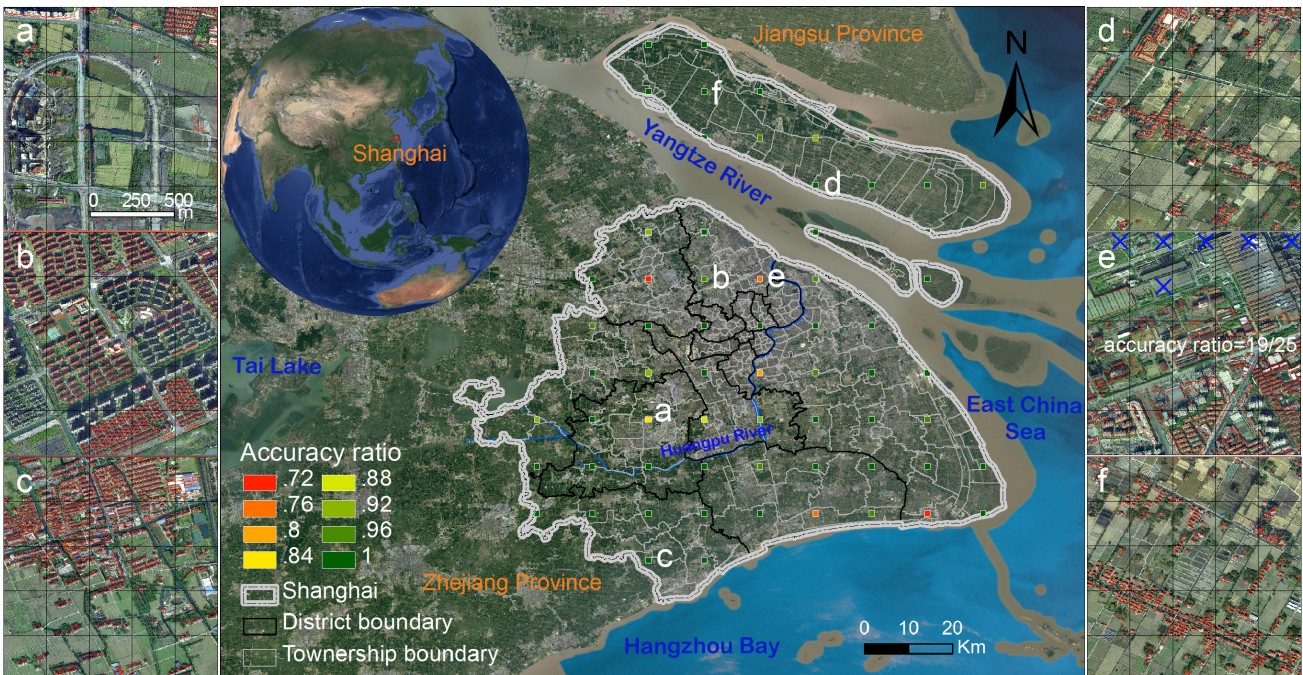

**Figure 1: Location of Shanghai, and comparison between building footprint (red polygon) and real aerial image (from Google Earth) by manual visual judgment.**



## 2.2 Outline of the methodology

To produce a high-resolution building asset value map for Shanghai, the following steps and data inputs are required (as shown in Figure 2): (1) statistical township floor area estimations by district-level combined with township population information, (2) disaggregation of floor area statistics from township-level to grid cell by ancillary data, i.e.,

building footprint map and the LandScan population density dataset, and model accuracy analysis by comparing the results with real statistical data, (3) building stock value distribution mapping by grid cell and floor area, multiplied by the replacement cost per unit floor area from field surveys and statistical data, (4) and for examining the feasibility of the building stock value map, a flood damage assessment is performed under an extreme flood scenario of Shanghai.

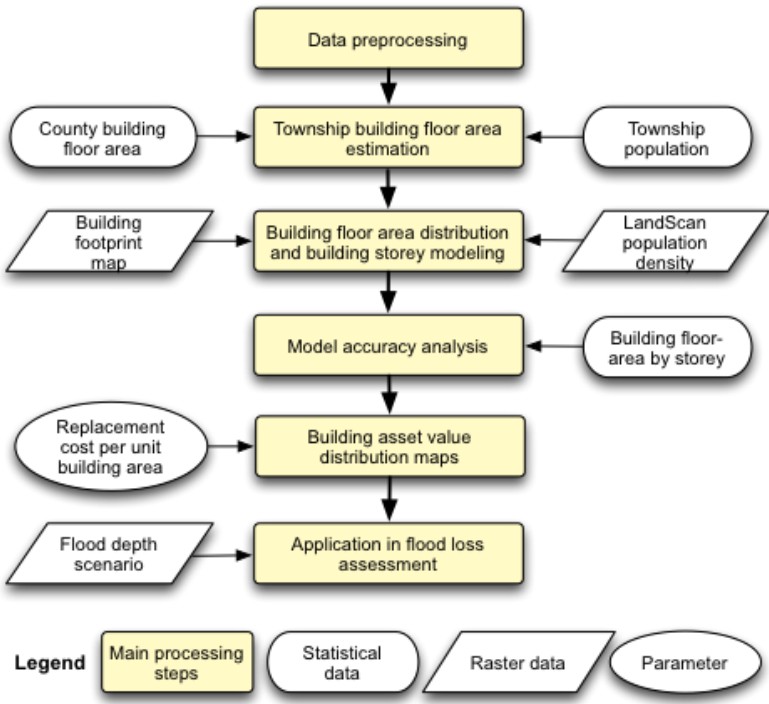

**Figure 2: Flowchart of the building asset value mapping methods.**

## 2.3. Input data and processing method

For producing a building stock value map of Shanghai, we first need to know where buildings are located. The Map World of Shanghai (www.shanghai-map.net) provides an electronic map with building footprints of Shanghai, generated from high-resolution remote sensing data of the Shanghai Municipal Institute of Surveying and Mapping

(SMSB) for the year 2013 and 2014 (see Table 1 for more details). For the overlay analysis, we transform this map from vector to raster format (i.e., 2.5m×2.5m).



To examine the accuracy of the building footprint map, we randomly select 60 places (approximately 1.25km×1.25km) to compare the building footprint map with the high-resolution (meter-level) remote sensing image from Google Earth. Following, each place is divided into 25 cells uniformly, to determine whether they are the same for each cell. This is done by performing a manual visual judgment. With this approach, we can determine the accuracy (*Accuracy ratio*) of the building footprint map for each place by,

$$Accuracy\ ratio\ =\ \mathrm{n}/25, \tag{1}$$

where $n$ indicates the number of cells of which the building footprint distribution is 50 percent or higher identical to the building distribution of the remote sensing image. The building footprint map provides a foundation (or base map) for further disaggregation of township-level floor area to grid cell.

**Table 1.** Introduction of the data used for building stock value mapping in Shanghai.

| Data name | Data types | Spatial resolution | Data source | Data year |
|---|---|---|---|---|
| Building footprint maps | Vector | Per-building | Map World of Shanghai: www.shanghai-map.net | 2013~2014 |
| Total population per township | Statistical data | Township-level | Shanghai Municipal Statistics Bureau | 2010 |
| Total population per district | Statistical data | District-level | Shanghai Municipal Statistics Bureau | 2014 |
| Total floor area with 6 floor-types | Statistical data | District-level | Shanghai Municipal Statistics Bureau | 2014 |
| LandScan population density | Raster | 30" (~800m) | Oak Ridge National Laboratory | 2010 |
| Township administrative boundary map | Vector | Township | National Science & Technology Infrastructure of China (www.geodata.cn) | 2002 |
| Flood inundation scenario data | Raster | 300m | Ke (2014) | None |

### 2.3.1 Township-level floor area estimation

SMSB (2015) provide data on the gross BFA both per building storey and district, but this data is not publicly available on the township-level for Shanghai. Fortunately, Township-level population census data is available from the *Tabulation on the 2010 population census of Shanghai Municipality* (2012). To estimate the building density, we use population density as a proxy, which is a common approach in dasymetric mapping (Aubrecht et al., 2013). We suppose that the per capita BFA is the same within a district, as the average land area for the 17 districts of Shanghai is only 373 square kilometers calculated from the statistical yearbook (SMSB, 2015), the spatial heterogeneity of per capita floor area is not obvious although it shows differences in larger districts like Pudong and Chongming. With this assumption, township-level BFA ($T_{area}$) can be estimated by,



$$T_{area} = POP_t \times \frac{C_{area}}{POP_c}$$

(2)

where $POP_t$ is the total population of a township, $C_{area}$ and $POP_c$ are, respectively, the statistical total BFA and the statistical total population of the district that the township belongs to (Table 2). As shown in Table 2, the total

5   population of Shanghai surpasses 24 million in 2014. The gross BFA of Shanghai reached 1153 km$^2$ and 53% of this BFA are residential buildings. Among the building types, buildings with 1-7 storeys (low-rise buildings) takes up to 68% of all floor area, buildings with storeys between 8 and 19 (medium-rise buildings) takes up to 21%, while only 11% of the BFA belongs to buildings with more than 20 storeys (high-rise buildings).

10   **Table 2.** District-level building floor area of Shanghai in 2014 (in km$^2$). Both subdivided in residential and non-residential and for building storey (and the total population in Shanghai of the year of 2014 living on these floors). The specified population is the total population for each district in 2014.

| District | Population (thousand) | Gross building floor area (km$^2$) | | | | | | | |
|---|---|---|---|---|---|---|---|---|---|
| | | Residential building | Non-residential building | 1~7 storey | 8~10 storey | 11~15 storey | 16~19 storey | 20~29 storey | >30 storey |
| Pudong | 5451.2 | 142.94 | 121.82 | 179.09 | 8.2 | 32.2 | 18.86 | 15.41 | 10.99 |
| Huangpu | 682 | 17.58 | 19.61 | 14.54 | 1.56 | 1.74 | 3.15 | 9.17 | 7.02 |
| Xuhui | 1109.7 | 33.94 | 25.29 | 31.79 | 2.36 | 4.71 | 5.97 | 9.92 | 4.47 |
| Changning | 698.6 | 24.11 | 15.6 | 19.81 | 1.95 | 3.51 | 2.99 | 6.93 | 4.52 |
| Jing'an | 248.6 | 8.19 | 9.42 | 5.52 | 0.61 | 0.66 | 1.39 | 5.56 | 3.87 |
| Putuo | 1296.1 | 35.99 | 22.18 | 31.32 | 1.49 | 5.57 | 5.86 | 9.68 | 4.24 |
| Zhabei | 848.5 | 22.61 | 14.94 | 22.28 | 1.37 | 4.18 | 2.57 | 4.81 | 2.35 |
| Hongkou | 838.2 | 22.29 | 13.17 | 19.46 | 1.24 | 2.4 | 3.72 | 5.75 | 2.89 |
| Yangpu | 1323.7 | 33.78 | 23.34 | 36.99 | 2.25 | 5.99 | 5.54 | 5.43 | 0.92 |
| Minhang | 2539.5 | 75.21 | 54.18 | 88.76 | 6.89 | 22.43 | 8.79 | 2.21 | 0.32 |
| Baoshan | 2024 | 54.02 | 37.75 | 68.84 | 2.55 | 10.6 | 7.21 | 2.09 | 0.46 |
| Jiading | 1566.2 | 33.64 | 40.41 | 54.4 | 1.92 | 7.13 | 7.28 | 3.06 | 0.28 |
| Jinshan | 797.1 | 14.66 | 25.44 | 35.35 | 0.79 | 2.66 | 0.97 | 0.31 | 0.02 |
| Songjiang | 1755.9 | 39.46 | 49.76 | 70.89 | 2.05 | 9.04 | 5.55 | 1.68 | 0.02 |
| Qingpu | 1208.3 | 21.24 | 31.11 | 44.39 | 0.92 | 3.01 | 3.47 | 0.55 | 0.01 |
| Fengxian | 1167.6 | 21.62 | 29.2 | 42.95 | 0.65 | 2.49 | 3.54 | 0.84 | 0.35 |
| Chongming | 701.6 | 9.66 | 9.28 | 17.87 | 0.27 | 0.73 | 0.06 | 0.02 | 0 |
| **Total** | **24256.8** | **610.94** | **542.5** | **784.25** | **37.07** | **119.05** | **86.92** | **83.42** | **42.73** |
| **Total (in %)** | | **53.0%** | **47.0%** | **68.0%** | **3.2%** | **10.3%** | **7.5%** | **7.2%** | **3.7%** |

Data source: SMSB (2015).


### 2.3.2 Building floor area distribution and building storey modeling

Based on both the building footprint map and the township-level BFA estimation, this paper uses population density (i.e., LandScan population grid) as ancillary data to represent the floor area density within a township. This implies that we assume that the floor area distribution is proportional to the population distribution within a township. It is not uncommon to assume that the floor area and population are closely related (Naroll, 1962). In this study, the Pearson correlation value for this relationship reached 0.99 for the 17 districts of Shanghai. Grid cell-level floor area ($F_{area}$) and the derived building storey ($S_{building}$) within a township can be calculated by,

$$F_{area} = T_{area} \times \frac{G_{pop}}{\sum G_{pop}},$$  (3)

$$S_{building} = \frac{F_{area}}{g_{area}},$$  (4)

where $G_{pop}$ is the LandScan population value in the grid cell (as shown in Figure 3), $g_{area}$ is the building occupied area of the related grid cell.

For validating the accuracy of the BFA distribution, first, as shown in Equation 4, for each grid-cell, building storey can be derived by dividing the estimated BFA with the area of the grid cell occupied (i.e., 2.5m×2.5m); then the modelled gross BFA can be calculated by storey types in each district; finally, the difference between the modelled BFA and the real statistical BFA (as shown in Table 2) by different storey (i.e., low-rise building, medium-rise building and high-rise building in this study) can be identified.

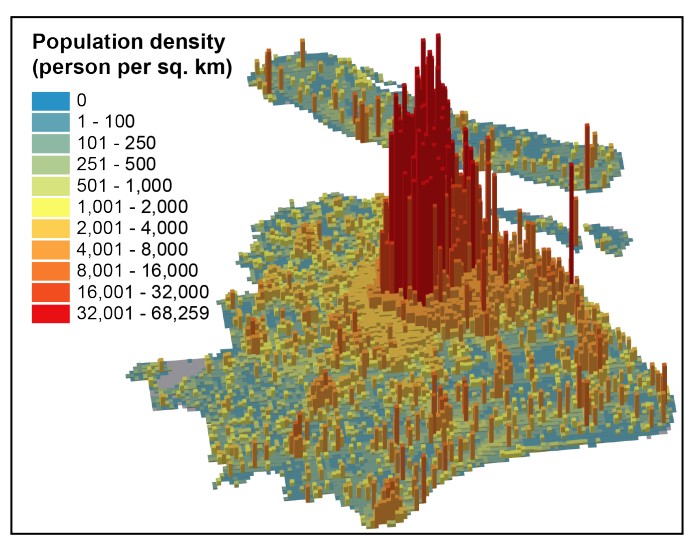



**Figure 3: LandScan population density of Shanghai in 2010.**

### 2.3.3 Valuation of building assets

In the research field of disaster loss estimation, there is an important distinction in methods for the valuation of assets
at risk. Researchers either use replacement costs or depreciated values within their loss calculation methods (Blong,
2003; Kreibich et al., 2010; Penning-Rowsell, 2010). Replacement cost can be interpreted as the costs of rebuilding a
property exactly as it was prior to the disaster, regardless of any depreciation due to its age. This approach provides
valuable information on potential pay-outs, and is therefore interesting from an insurance perspective (Jongman et al.,
2012). The depreciated value can be interpreted as the remaining property value, corrected for its depreciation by age;
the depreciated value is often assumed to be the true costs associated with the loss of the asset (Penning-Rowsell,
2010). Important to note is that the depreciation value is lower than the replacement value. Given that flood insurance
is still not well developed in China, especially in rural areas, governmental financial support guarantees the
post-disaster reconstruction of the house of the inhabitant. As such, this paper uses the unit cost (i.e. the construction
cost excluding the cost of land, per square meter, from field survey) of replacement of the building to produce a
building stock value map for disaster loss estimation of Shanghai.

**Table 3.** Overview of construction cost per square meter for the most common building types.

| Building type | Construction cost per m$^2$ |
|---|---|
| Residential buildings | 1100-5900 CNY (US$165-884) |
| Commercial buildings | 1560-14400 CNY (US$234-2158) |
| Industrial buildings | 900-9000 CNY (US$135-1349) |
| Public buildings | 1500-7320 CNY (US$225-1097) |

Because the individual costs per building are impossible to obtain, we use the mean construction costs by building
storey for calculating the building stock value. A standard of construction costs for different types of works in
Shanghai was given in Chinese Yuan (CNY) per square meter of floor area by the Shanghai Municipal Housing, Land
and Resources Administration Bureau (SHLRAB, 2007). Table 3 provides an overview of the construction costs for
different building types. Because there is no information available on the relative share of residential, commercial,
industrial and public buildings per urban land-use cell within Shanghai, we use building storey-based construction cost
for estimating the building asset value. This means that for low and medium storey buildings (1~19 storey), we assume
construction costs of approximately 3230 CNY per square meter, while for high-storey buildings (over 19 storey), we
assume construction costs of 6750 CNY per square meter. This allows determining the building stock value (*Bv*) in



each grid cell, based on the modelled BFA and unit cost ($Cost_{uint}$) of construction for the different building storeys,

$$Bv = F_{area} \times Cost_{uint}. \tag{5}$$

## 3. Results

### 3.1. Building footprint distribution accuracy

For examining the spatial distribution of the building footprint map from SMSB, 60 places (1.25km×1.25km) were uniformly chosen for distribution comparison between building footprint and aerial images (as shown in Figure 1 and explained in section 2.2). Figure 1e shows a calculation example of the accuracy (Blue cross marks show the parts where the accuracy is zero). The Figure presents six representative pairs of the data, where the red polygons are the building footprint boundary. The accuracy results show that the average accuracy ratio is 0.97 for the 60 places, with

lowest accuracy estimates of only 0.72 in two places (Figure 1). This means that this building footprint map can represent the distribution of the real buildings very well. Compared to the land-use map, the vector-based building footprint map can be rasterized to any spatial resolution to smoothly facilitate the overlay analysis with hazard intensity for natural disaster loss estimation.

### 3.2. Building asset value mapping

Based on the statistical data of both the district-level BFA (Table 2) and township-level population, township-level BFA can be estimated under the assumption that per capita BFA is the same within a district. Township-level BFA ratio can be calculated (as shown in Figure 4) by BFA divided by total area for each township. As expected, the BFA ratio in the downtown area is greater compared to the surrounding area (see Figure 4). More specifically, this means that the average relative building storey is greater in Huangpu, Xuhui, Jingan, Putuo, Zhabei and Hongkou, compared to the sub-urban areas

of Shanghai. Based on the township-level BFA estimation, combined with the building footprint map and the LandScan population density, grid cell-level BFA and building storey can be derived from Equation 3 and Equation 4, respectively.

   As Figure 5 shows, there is a significant relationship between derived BFA and real statistical BFA by three-type buildings at district level, and the Pearson correlation coefficient reached 0.93. The figure also shows that most of the district's

medium-rise BFA is slightly overestimated, whereas both low-rise and high-rise BFA are slightly underestimated for most of the districts. The spatial resolution of the LandScan population density data is approximately 800m for Shanghai. This coarse resolution smooths out the high- and low- population density area compared to a 2.5m spatial resolution of the building footprint map that is used here. This may be the main reason for the overestimation of medium-rise BFA in most of the districts of Shanghai. Overall, the modeled BFA distribution for different storeys reflects the real situation well in the district

scale to a certain extent as described above.




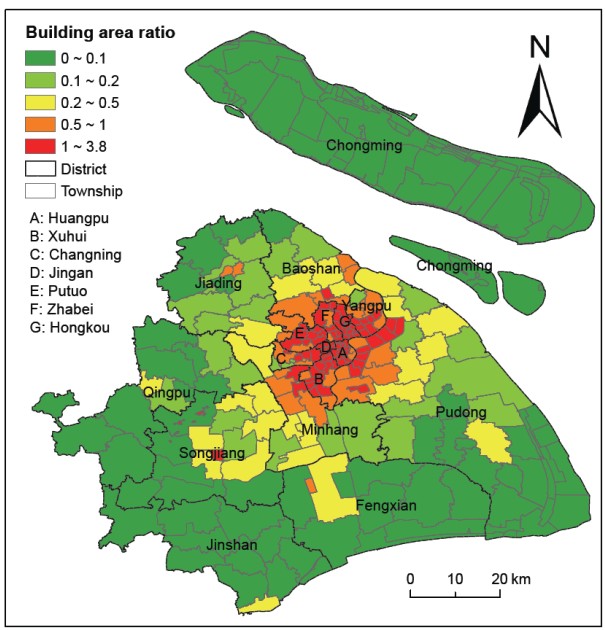

**Figure 4: Distribution of township-level building floor area ratio in Shanghai. Township-level BFA ratio is equal to BFA divided by total area for each township, BFA ratio reflects the virtual average building storey of the township (i.e., the building density distribution), BFA ratio surpass one indicates that the BFA is greater than the total land area of the township, i.e., multiple storeys buildings were dominant in this township.**

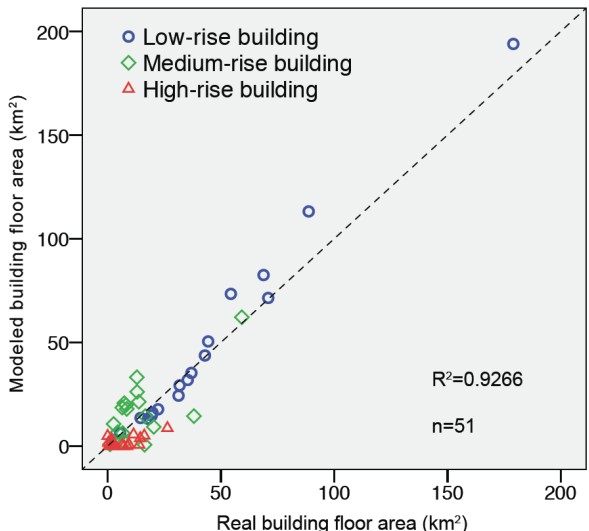

**Figure 5: Relationship between modeled building floor area and real statistical floor area for different storey building. Low-rise buildings: 1~7 storey, medium-rise buildings: 8~19 storey; high-rise buildings: over 20 storey.**



Finally building asset value can be calculated by BFA multiplied by unit per capita replacement cost in each grid cell that is subject to building storey as described above (as shown in Figure 6). Total replacement building value reached 3856.6 billion CNY (about US$ 578.3 billion) for Shanghai in 2014.

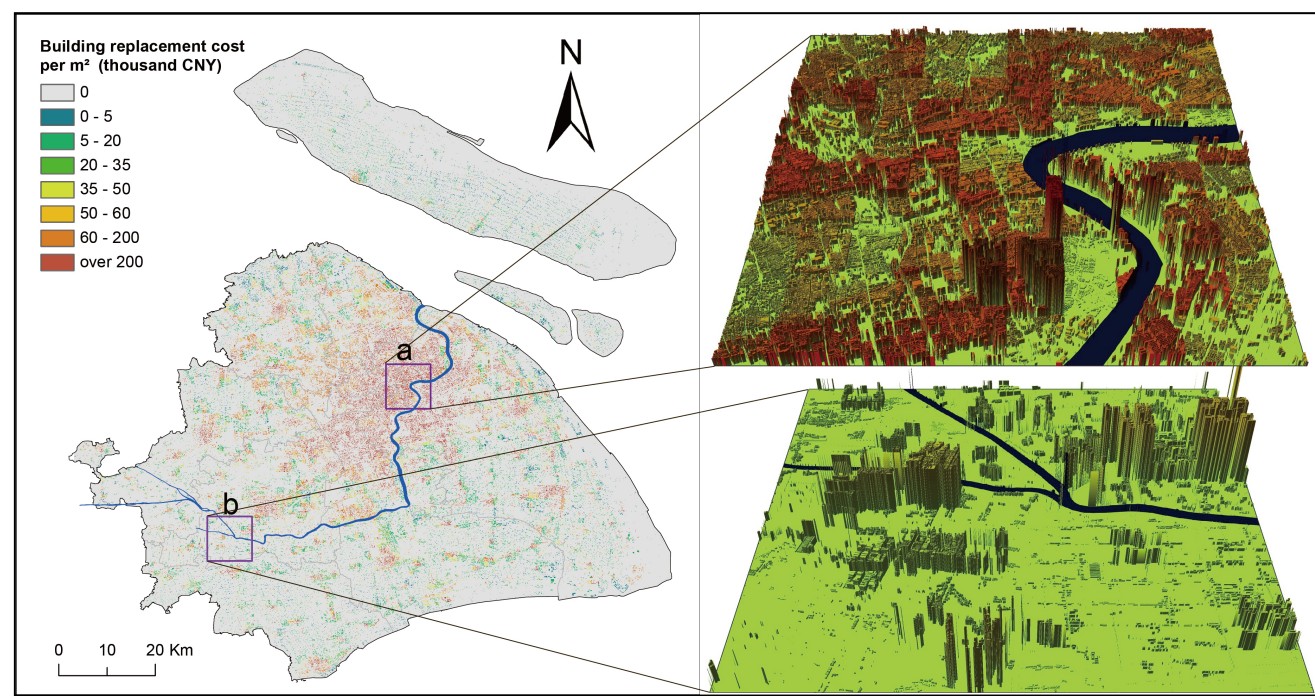

**Figure 6: Distribution of disaggregated building asset value (replacement costs in current price of 2014, grid cell 2.5m×2.5m) in Shanghai, and two regions are zoomed in to show the building asset value density in 3D view.**

## 4. Discussion

10  **4.1 Application of building asset value map in the scenario-based flood damage modelling**

The most commonly used method to assess flood losses is the use of a combination of depth-damage curves, an exposure map and an inundation map. By using these depth-damage curves, information on inundation and the spatial distribution of people and assets (often in the form of a land-use map) can be combined to assess the damage for any given cell on the exposure map, based on the depth in the inundation map (Jongman et al., 2012; Koks et al., 2014;

15  Muis et al., 2015). Figure 7 shows the application diagram of the building asset value map used in flood loss modelling.


Both flood inundation map (Figure 7b) and depth-damage curves (Figure 7c) were from Ke (2014), who modelled a flood with a 1/10,000yr probability in the Huangpu River of Shanghai, assuming no flood protection (Figure 7b). According to Ke (2014), commercial buildings has the highest vulnerability to floods than other building types, here we use the depth-damage curve of commercial buildings (Figure 7c) for a simple modelling of flood damage. As

5    depth-damage curve is recognized as the primary source of uncertainty in flood damage estimation (de Moel and Aerts, 2011), one should be cautious with the modelled flood damage here.

For building assets value at risk of flooding, in reality, however, only a small proportion of the building will be exposed to the flood. This is especially the case for high-rise buildings. Green (2010) identified a relationship between

10    population density and the ratio of net assets exposed to flooding and recommends that a fraction of $1/6^{th}$ of the assets should be considered as the exposed value when the population density is above 15000/km$^2$, a fraction of $1/4^{th}$ for a population density between 8000/km$^2$ and 15000/km$^2$, 1/2 for a population density between 1000/km$^2$ and 8000/km$^2$ and 1 for a population density under 1000/km$^2$ (Ke, 2014). Based on this assumption, we use the LandScan population density and building value map to determine the real exposed building asset value.

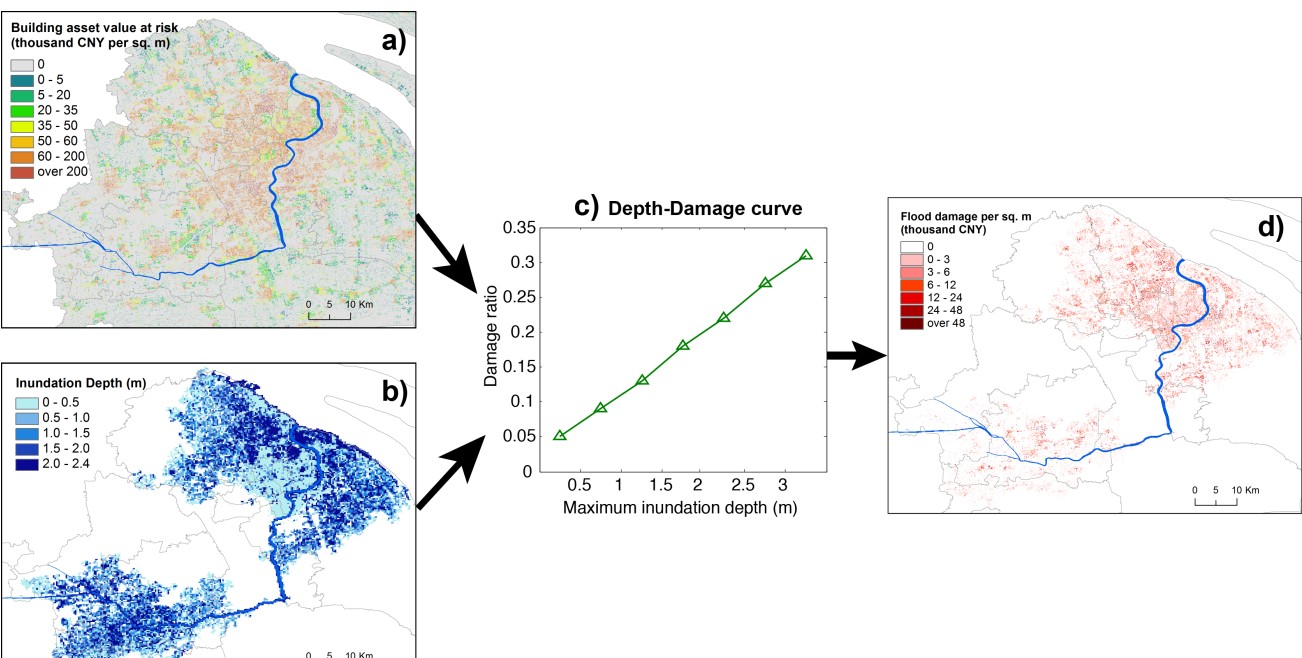

**Figure 7: Flow diagram of the flood damage modelling methodology based on the maps of building asset value at risk (a) and inundation depth (b), and the depth-damage curve (c). Both flood inundation map (under the scenarios of no embankment along the Huangpu River with a probability of 1/10,000 chance of occurrence) and depth-damage curve**



**of Shanghai are from Ke (2014).**

Finally, we can calculate the real exposed building asset value reached 665.9 billion CNY (about US$ 99.8 billion) by overlay analysis of Figure 7a and Figure 7b, which takes up 38.4% of the total building asset value of Shanghai. Combined with depth-damage curve (Figure 7c), the total direct economic damage for buildings further can be are calculated with the sum of the damages in all inundated grid cells (Figure 7d), which comes to a total of approximately 75.2 billion CNY (US$ 11.3 billion) within this flood scenario.

## 4.2 Uncertainties

Uncertainties arise from several sources. For building asset value estimation, the building footprint map from Map World of Shanghai is the most precise building distribution information we can acquire, with a sampling precision of up to 97%. Unfortunately, there is no building storey attribute for the building footprint map. To produce a more realistic building asset value map, this paper resorted to use the LandScan population density data for deriving the building storey distribution information under the assumption that per capita BFA is the same within a township. Due to the coarse spatial resolution of the LandScan population density dataset (about 800m in Shanghai), both the high- and low-rise BFA was underestimated, whereas medium-rise BFA in most districts was overestimated (as shown in Figure 5).

Building construction cost depends on the type of building (i.e., building use type, architectural structure and building storey). Due to the shortage of such information, this paper use present average unit costs of construction, according to the derived building storey as described above. By using the average unit cost of construction for 1-19 storey buildings, the total building asset value may be overestimated, as the low-rise BFA take up 68% of the total BFA for Shanghai, and the proportion of BFA is only 11% for over 20-storey building (Table 2).

Capturing the value of building asset is an important and integral part of any natural disaster risk assessment modeling effort. The term "*building replacement value*" is being used in the disaster loss estimation field (i.e., for China), which is the cost to rebuild a property exactly the same as pre-disaster. However, the real costs for rebuilding can be higher because the quality of the new replaced properties is usually new and safer. In other words, the overestimation of the building asset value, using present construction cost, may have practical implications for disaster risk management.

As far as we know, there is no other available building asset value map for Shanghai for comparison, but the open Global Exposure database of Global Assessment Report on Disaster Risk Reduction 2015(GEG15) (De Bono and



Bruno, 2015) is a few available sources that can provide capital stock value (in 2014) distribution map (grids by 5km ×5km) of China, including Shanghai, and the total capital stock reached US$ 1190.3 billion in 2014, which means that the building asset value (US$ 578.3 billion as described above) accounts for 48.6% of the total capital stock of Shanghai. Unfortunately, we cannot compare the difference of building asset value distribution between GEG15 and this study, as GEG15 did not differentiate building asset value from total capital stock.

The biggest advantage of the disaggregation method developed in this paper is that you can easily produce a building asset value map for local natural disaster risk assessment if three main datasets are available, i.e., a footprint map, a population density map, and census statistical building floor area data. While the three datasets are available all over China as described above. As such, the method used for building asset value mapping is applicable for other regions of China, even for other countries if the three main datasets are available.

## 5. Conclusions

For flood risk management, one of the core questions is how much flood damage would be induced if an area were flooded. For good estimates, building stock information is extremely valuable. Unfortunately, building stock data often only exists on an administrative scale, which is inconsistent with the distribution of the real hazard extent. As such, this mismatch raised an urgent need for seeking new techniques to produce a geo-referenced building asset value map to overlay with the hazard intensity. This paper outlined a methodology to produce a high-resolution (2.5m×2.5m) building asset value map, using the best data currently available for its implementation, for city-scale disaster risk assessment. The model is capable of identifying the spatial distribution and the density of building asset value. As such, this estimated map and its methodology can be of great value for flood risk modeling, and can be used in the analysis of other disasters as well. The exposure modeling methodology presented combines and provides original perspectives to assess exposed building asset value at risk of China. The dataset produced would be instrumental in providing key baseline data and information for natural disaster risk management of Shanghai; moreover, this methodology is also portable to be used in other cities of China as the building footprint map is constructing allover China.

**Acknowledgements**:This study was financially supported by the National Key Research and Development Program—Global Change and Mitigation Project "*Global change risk of population and economic system: mechanism and assessment*" under Grant No. 2016YFA0602403 and the National Natural Science Foundation of China under Grant No. 41571492. The authors are grateful to Dr. Qian Ke for providing the flood hazard scenario data of Shanghai.





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
