# Peer review of "Building Asset Value Mapping in Support of Flood Risk Assessment: A Case Study of Shanghai, China"

_Natural Hazards and Earth System Sciences, 2017_

## Referee Comment (RC1) · Anonymous Referee #1 · 20 Mar 2017

The discussion paper entitled 'Building Asset Value Mapping in Support of Flood Risk Assessment: A Case Study of Shanghai, China' effectively developed a methodology to map the building asset value using Shanghai as a case study and further applied the results in a flood damage estimation under a flood scenario of Shanghai to testify the flexibility of the BFA map. But some problems however may affect the accurate evaluation. In specific:

1): LandScan population (2010) was used as ancillary data together with building footprint map and the township-level BFA estimation (2014) to represent the floor area density within a township. However, the population of the city of Shanghai increased by 1 million during 2010 to 2014. The time inconsistency should be fixed or at least

discussed (P5, Table1);

2): While the floor area and population are generally closely related and high correlation between them in district level has been presented, to use population data as proxy to estimate the building density can still bring major errors in some areas of Chinese city, such as villa residential and 'village in the city'. For these areas, the building density for the same number of population should be totally different. For these part, a correlation analysis with validation with random point instead of district level should be necessary, especially taking account for the high-resolution results in this research (P5, L15∼20);

3): The explanation of the estimation of construction costs is not clear. The basis of 3230 CNY for medium story and 6750 CNY for high story is not convincing enough. Besides, spatial differences are neglected with the mean value (P8, L24∼27);

Minor problem: The sentence 'both the high- and low-rise BFA was underestimated' should be 'both the high-rise and low-rise BFA was underestimated' (P13, L14).

---

## Referee Comment (RC2) · Anonymous Referee #2 · 23 Mar 2017

The paper presents a method to disaggregate (i.e., downscale) building assets values from census information at the district level to a finer resolution (grid of 2.5 x 2.5 m) using information on population distribution (from LandScan, resolution of ∼800m) and building footprint maps. As a case study, the proposed method is applied to the City of Shanghai (China). The topic is relevant and certainly meets the potential interest of the readers of NHESS. However, I have major concerns with the method proposed and with some of its basic assumptions. Moreover, the presentation of the paper is too lacking to allow a deep and clear understanding of both the scientific significance and the novelty of the work. For these reasons, I suggest rejection and resubmission of a new, improved paper.

The points that need to be considerably improved are detailed in the following, along with some minor comments.

Major points

1) The paper need a thorough English revision, which now is not acceptable for publication. Lot of typos, awkward and incorrect sentences are present in the text, and should be fixed preferably with the help of a native speaker.

2) The use of terminology needs much care. Clear definitions should be given before describing the method. E.g., does the buildings floor area (BFA) from census refer to a single storey or to all the storeys of a building? Moreover, the procedure should be better outlined introducing clear notation and using a suitable number of equations. Importantly, the variables in every equations should be unequivocally linked to the quantities referred to in the text. E.g., I really did not understand what variable, in the equations shown, denotes the BFA.

3) The description of the overall method is very tangled, and it should be made considerably clearer. Specifically, the description of available data, of their use, and of the overall method are all mixed together, in a way that it is very difficult to grasp a clear understanding of the method proposed by the Authors.

4) The method used to evaluate of the accuracy of the building footprints is dubious. For example, based on the definition of the accuracy ratio (Eq. 1), if the correspondence between vector building footprints and aerial images is equal to 51% in each one of the 25 cells (i.e., a total discrepancy of 49%), the accuracy ratio for the entire place is equal to 1! Moreover, Fig. 1 suggests that the building area from vector building footprint always underestimates the building area provided by the aerial images.

5) I have concerns about the resolution of the final grid (2.5 x 2.5 m) given the resolution of the input data. The buildings footprints are the only data comparable in terms of resolution (as they are provided in vector form), whereas the assumption of invariant

distributions at finer scales are incompatible with the final resolution of the grid, as clearly explained and demonstrated by, e.g., Figueiredo and Martina (2016).

6) Sec. 2.3.3 "Valuation of building assets": the Authors refers to reconstruction costs and to construction costs without a clear distinction of the two. Moreover, flooding of nearly flat urban areas is unlikely to cause the reconstruction of buildings (i.e., walls, etc., which are accounted for in the construction cost). Rather, major economic losses are linked to damage of goods. Moreover, damages are likely to affect only the ground floor, so I don't understand the reason why high-storey buildings should have a loss per unit area that is more than twice that of a low-storey building.

7) Evaluation of the proposed method. Fig.5 shows a comparison between modeled and real statistical building floor area (BFA). Despite the fact that the comparison is carried out at the district level (i.e., at a very coarse resolution), substantial discrepancies are shown, particularly for small values of BFA. Why?

8) The flooding scenario used in the final application of the exposure dataset in order to produce estimates of economic losses is far from reality. The return period (10'000 years) is extremely high if compared with common return periods used in the engineering practice. The text says that no flood protection are assumed in the modeling of flooding. It is not clear if flood protections (e.g., levee) actually exist but they have been disregarded in the calculation of the flooding scenario, or if they do not exist (if this is the case, why saying "assuming no flood protection"?).

Minor points

1) "Methodology" is not correct in this context; authors should use "method" instead. "Methodology" pertains to the "study of methods". This issue applies to paper's entire text.

2) Authors should pay attention to properly define abbreviations. E.g., "LULC" is never defined; "BFA" is defined only in the Abstract, and should be redefined when first used

within the paper (i.e., at page 2, line 20).

3) Abstract. What does the "immense analytical flexibility" refer to? To the usefulness of the exposure dataset? It seems to me an overstatement.

4) Pag. 2, l 2-3: "The main cause for this uncertainty is ... of water inundation depth" could be supported by additional references, e.g. "...of water inundation depth, both in urban and rural areas (e.g., Apel et al., 2016 and Viero et al., 2014, respectively)".

5) Fig. 1: a horizontal line is missing in panels b and e, respectively. Please consider increasing the thickness of the grid lines in the lateral panels. Please control "township".

6) Sec. 2.2: Point (4) is not devoted to produce the high resolution building asset value map, as it concerns the test for the application of the final result of the proposed method.

7) The names of the variables that appear in the (few) equations are very little informative, and should be chosen with greater care. In many cases, the names of variables should be exchanged with the subscript. Indeed, the name of the variable commonly identifies the kind of measure (e.g., A for areas, C for costs, and so on...), and the subscripts should provide further specifications (e.g., the level at which the area refer to, such as district or township).

8) Eq. (3): ' is not defined. What does the summation refer to?

9) Eq. 5: unit, not uint.

Additional references

Apel, H., Martìnez Trepat, O., Hung, N. N., Chinh, D. T., Merz, B., Dung, N. V., 2016. Combined fluvial and pluvial urban flood hazard analysis: concept development and application to Can Tho city, Mekong Delta, Vietnam, Nat. Hazards Earth Syst. Sci., 16, 941–961, doi:10.5194/nhess-16-941-2016.

[Figure]

Viero, D. P., Peruzzo, P., Carniello, L., Defina, A., 2016. Integrated mathematical modeling of hydrological and hydrodynamic response to rainfall events in rural lowland catchments, Water Resources Research, 50(7), 5941–5957, doi:10.1002/2013WR014293.

---

## Author Comment (AC1) · 26 Apr 2017

We thank the referee 1 for his/her careful reading of the paper, and we have found the comments extremely useful while working on a new draft of the paper with substantial changes. We have adopted the recommendations from referee 1.

Response to anonymous Referee #1

The discussion paper entitled 'Building Asset Value Mapping in Support of Flood Risk Assessment: A Case Study of Shanghai, China' effectively developed a methodology to map the building asset value using Shanghai as a case study and further applied the results in a flood damage estimation under a flood scenario of Shanghai to testify

the flexibility of the BFA map. But some problems however may affect the accurate evaluation. In specific:

1): LandScan population (2010) was used as ancillary data together with building footprint map and the township-level BFA estimation (2014) to represent the floor area density within a township. However, the population of the city of Shanghai increased by 1 million during 2010 to 2014. The time inconsistency should be fixed or at least discussed (P5, Table1);

Response: Thanks for indicating the data limitation problem in this study. Because we did not acquire LandScan 2014, we have to resort to LandScan 2010 that we had. We totally agree with your suggestion, and will add a discussion for this issue in the Discussion section.

2): While the floor area and population are generally closely related and high correlation between them in district level has been presented, to use population data as proxy to estimate the building density can still bring major errors in some areas of Chinese city, such as villa residential and 'village in the city'. For these areas, the building density for the same number of population should be totally different. For these part, a correlation analysis with validation with random point instead of district level should be necessary, especially taking account for the high-resolution results in this research (P5, L15_20);

Response: Thanks for your constructive comments. As you referred, the population density is not perfectly correlated with building area, such as villa residential and 'village in the city', the spatial heterogeneity for this correlation should be even bigger in finer spatial scale. We totally agree with your suggestion, and did a finer spatial scale validation: firstly, we acquired a real building distribution vector layer for downtown area of Shanghai, which including the building height information, according to the relationship between number of storeys and building height sampling, we transformed building height to building storeys, then we can calculate building floor area in each regular grid

cell; b) after that, we can compare the modeled building floor area with the real building floor area as you suggested and as a necessary supplement of validation except for district level correlation analysis.

Detailed revisions are as follows: 1) we add a new figure in the main text (Section 3) to show the validation of modeled building floor area in the downtown area of Shanghai. 2) we add a paragraph to describe the validation process and results in the new Section 3.2. "As population density is not fully correlated with BFA, such as villa residential and 'village in the city' in some city of China, for further validating the accuracy of modeled BFA in sub-district level, we compared the modeled BFA with real BFA. As Figure 6 shows, . . . . . ."

Figure 6: Building floor area validation by different grid cell size in part of the downtown area of Shanghai. a) Real building height in the downtown area of Shanghai (i.e., the sampling area for validation). b) Relationship between building height and number of storeys in Shanghai acquired by random building surveys. c) Real building floor area calculated by number of storeys via a) and b) multiplied with occupied area of the building footprint. d) Modeled building floor area as described above in this article. e) Distribution of relative error between modeled BFA and Real BFA by five different grid cell size. f) Relationship between modeled BFA and Real BFA for regular grid cell size of 1600 m. g) Spatial relative error distribution between modeled BFA and Real BFA for regular grid cell size of 1600 m.

3): The explanation of the estimation of construction costs is not clear. The basis of 3230 CNY for medium story and 6750 CNY for high story is not convincing enough. Besides, spatial differences are neglected with the mean value (P8, L24_27);

Response: Thanks for your comments. From the perspective of reconstruction cost needed post-disaster, we use replacement cost to estimate the building asset value according to building types by reference to "Construction cost standard of Shanghai". We did not consider the spatial differences of replacement cost across Shanghai (with

a total area of 6340 square kilometers in size) because of data issues even though this differences are actually exist. According to your comments, we consider to revise this part as follows: a) explain the replacement cost parameter more clearly in the main text; 2) add a discussion for this limitation in the Discussion section.

4): Minor problem: The sentence 'both the high- and low-rise BFA was underestimated' should be 'both the high-rise and low-rise BFA was underestimated' (P13, L14).

Response: Thanks for your reminding. We have adopted your suggestion. We will check the English language issues in the revised manuscript.

Thank you again for your comments. These comments are extremely helpful for us.

Please also note the supplement to this comment:
http://www.nat-hazards-earth-syst-sci-discuss.net/nhess-2017-17/nhess-2017-17-AC1-supplement.pdf

---

## Author Comment (AC2) · 26 Apr 2017

We thank the referee 2 for his/her careful reading of the paper, and we have found the comments extremely useful while working on a new draft of the paper with substantial changes. We have adopted the recommendations from referee 2.

Response to anonymous Referee #2

The paper presents a method to disaggregate (i.e., downscale) building assets values from census information at the district level to a finer resolution (grid of 2.5 × 2.5 m) using information on population distribution (from LandScan, resolution of ∼ 800m) and building footprint maps. As a case study, the proposed method is applied to the City

of Shanghai (China). The topic is relevant and certainly meets the potential interest of the readers of NHESS. However, I have major concerns with the method proposed and with some of its basic assumptions. Moreover, the presentation of the paper is too lacking to allow a deep and clear understanding of both the scientific significance and the novelty of the work. For these reasons, I suggest rejection and resubmission of a new, improved paper.

The points that need to be considerably improved are detailed in the following, along with some minor comments.

Response: Thanks for your comments, we will have responses for each comment as follows.

Major points 1) The paper need a thorough English revision, which now is not acceptable for publication. Lot of typos, awkward and incorrect sentences are present in the text, and should be fixed preferably with the help of a native speaker.

Response: Thanks for your advice, we will check the English language issues in the revised manuscript, and will be looking for partners whose native language is English to help modify the language problem, or using English language editing service.

2) The use of terminology needs much care. Clear definitions should be given before describing the method. E.g., does the buildings floor area (BFA) from census refer to a single storey or to all the storeys of a building? Moreover, the procedure should be better outlined introducing clear notation and using a suitable number of equations. Importantly, the variables in every equation should be unequivocally linked to the quantities referred to in the text. E.g., I really did not understand what variable, in the equations shown, denotes the BFA.

Response: Thanks for your suggestion. We have adopted your suggestion: a) adding clear definition of special terminology including building floor area (BFA) and floor area ratio (FAR), "BFA refer to the total floor area inside the building envelope, including the

external walls, and excluding the roof, floor area ratio (FAR) is the ratio of a building's total floor area (BFA) to the size of the piece of land upon which it is built." b) we will improve the expression in the methodology section, including modify the names of the variables that appear in the equations, increase explanations about the subscripts. Thanks for your reminding.

3) The description of the overall method is very tangled, and it should be made considerably clearer. Specifically, the description of available data, of their use, and of the overall method are all mixed together, in a way that it is very difficult to grasp a clear understanding of the method proposed by the Authors.

Response: We totally agree with your suggestion. For resolving the confusion that both data and method are all mixed together that you referred. We will add a separate section, i.e., Section 2.2 Data, to explain clearly the description of available data (listed in Table 1), and of their use. Moreover, following by Section 2.3 Method for this study.

4) The method used to evaluate of the accuracy of the building footprints is dubious. For example, based on the definition of the accuracy ratio (Eq. 1), if the correspondence between vector building footprints and aerial images is equal to 51% in each one of the 25 cells (i.e., a total discrepancy of 49%), the accuracy ratio for the entire place is equal to 1! Moreover, Fig. 1 suggests that the building area from vector building footprint always underestimates the building area provided by the aerial images.

Response: Thanks for your comments on precision validation of building footprints data. On the one hand, this manuscript mainly use artificial vision judgement to decide the accuracy ratio of cells, which is a little subjective. However, under no better quantitative method for selection, artificial vision judgement is reasonable on precision validation of building footprints data. On the other hand, we totally agree with your reminding that the building area from vector building footprint always underestimates the building area provided by the aerial images, and we will add this conclusion to the main text for building footprint precision validation. Thanks again for your suggestion.

5) I have concerns about the resolution of the final grid (2.5 x 2.5 m) given the resolution of the input data. The buildings footprints are the only data comparable in terms of resolution (as they are provided in vector form), whereas the assumption of invariant distributions at finer scales are incompatible with the final resolution of the grid, as clearly explained and demonstrated by, e.g., Figueiredo and Martina (2016).

Response: We totally agree with your concerns about data resolution. Firstly, data resolution is always the obstacle for flood risk modelling especially for developing countries (such as China), so we resort to use high-resolution building footprints data (i.e., rasterized to 2.5 x 2.5 m), which provide the exact location of building distribution. However, due to data limitations, we have to assume invariant distributions within about 800 x 800 m extent when disaggregated building asset value to grid cell (using LandScan population density as ancillary data). In other words, the real resolution of disaggregated building asset value map is no more than 800 x 800 m, we just shown in 2.5 x 2.5 for highlighting the building distribution, as Figueiredo and Martina (2016) indicated "lower resolutions of the exposure model will in general lead to an overestimation of the affected buildings" in flood exposure estimation, this is also our intention for this study, i.e., improve the exposure data quality in natural disaster risk estimation. We will add a discussion for this limitation in the Discussion section.

6) Sec. 2.3.3 "Valuation of building assets": the Authors refers to reconstruction costs and to construction costs without a clear distinction of the two. Moreover, flooding of nearly flat urban areas is unlikely to cause the reconstruction of buildings (i.e., walls, etc., which are accounted for in the construction cost). Rather, major economic losses are linked to damage of goods. Moreover, damages are likely to affect only the ground floor, so I don't understand the reason why high-storey buildings should have a loss per unit area that is more than twice that of a low-storey building.

Response: Thanks for pointing out the uncertainty on simulation of flood damage scenario. We totally agree with your concerns. Firstly, for Sec. 2.3.3 Valuation of building assets, which aims at producing building assets value map for natural disaster risk assessment, such as earthquakes, tropical cyclone, sea level rise, and also floods. So, according to your comments, we consider change the title from "Building Asset Value Mapping in Support of Flood Risk Assessment ..." into "Building Asset Value Mapping in Support of Natural Disaster Risk Assessment ...". We will use "reconstruction costs" for the valuation of building assets according to your reminding. Meanwhile, as for the complexity of flood loss modelling, we consider deleted such content as it is not the main focus of this study, and we just want to show how flexible of the disaggregated building asset value map can be used in exposure assessment of floods in the Discussion section, but do not simulate the damages.

7) Evaluation of the proposed method. Fig.5 shows a comparison between modeled and real statistical building floor area (BFA). Despite the fact that the comparison is carried out at the district level (i.e., at a very coarse resolution), substantial discrepancies are shown, particularly for small values of BFA. Why?

Response: Thanks for your comments. Firstly, as shown in Figure 5 and Table 2 in the manuscript, most of the district's total BFA is lower than 50 km2. For each district in each building category (i.e., low-, medium- or high-rise building), we compare the census statistical value and modeled value (shown in Figure 5). As explained in the manuscript, "The figure also shows that most of the district's medium-rise BFA is slightly overestimated, whereas both low-rise and high-rise BFA are slightly underestimated for most of the districts. The spatial resolution of the LandScan population density data is approximately 800m for Shanghai. This coarse resolution smooths out the high- and low- population density area compared to a 2.5m spatial resolution of the building footprint map that is used here. This may be the main reason for the overestimation of medium-rise BFA in most of the districts of Shanghai."

Moreover, the population density is not perfectly correlated with building floor area, such as villa residential and 'village in the city', the spatial heterogeneity for this correlation should be even bigger in finer spatial scale. Substantial discrepancies are higher when the spatial scale is small (i.e., small values of BFA) than that of large spatial

extent (as shown in Figure 6e below, which will be added in the revised manuscript).

Figure 6: Building floor area validation by different grid cell size in part of the downtown area of Shanghai. a) Real building height in the downtown area of Shanghai (i.e., the sampling area for validation). b) Relationship between building height and number of storeys in Shanghai acquired by random building surveys. c) Real building floor area calculated by number of storeys via a) and b) multiplied with occupied area of the building footprint. d) Modeled building floor area as described above in this article. e) Distribution of relative error between modeled BFA and Real BFA by five different grid cell size. f) Relationship between modeled BFA and Real BFA for regular grid cell size of 1600 m. g) Spatial relative error distribution between modeled BFA and Real BFA for regular grid cell size of 1600 m.

8) The flooding scenario used in the final application of the exposure dataset in order to produce estimates of economic losses is far from reality. The return period (10'000 years) is extremely high if compared with common return periods used in the engineering practice. The text says that no flood protection are assumed in the modeling of flooding. It is not clear if flood protections (e.g., levee) actually exist but they have been disregarded in the calculation of the flooding scenario, or if they do not exist (if this is the case, why saying "assuming no flood protection"?).

Response: Thanks for pointing out the problem on simulation of flood damage scenario. Firstly, for Shanghai, flood protections (e.g., levee) actually exist but they have been disregarded in the calculation of the flooding scenario (as used in this study). We just want to show how flexible of the disaggregated building asset value map can be used in exposure assessment of natural disasters. Because its high complexity in flood risk assessment, there are large uncertainty for flood risk estimation, for example, for flood return period estimation and also its use in the engineering practice, much attention has been paid to univariate flood frequency analysis. However, flood risk in Shanghai is a multivariate process than depends on several random variables, such as sea level rise, tide, tropical cyclone, floods in the Upper Yangtze River. As we had used

bivariate joint probability distribution for the return period analysis of dust storms (Li et al., Risk Analysis, 33(1), 2013), the results indicated that an 81-year return period dust storm (when) using univariate will change to a 4-year return period dust storm (when) using bivariate analysis (Copula-based modelling). Considering the man-made reasons and other unexpected factors, even with good flood protection measures, it also induced large economic losses (as seen in New Orleans after Hurricane Katrina, and New York after Hurricane Sandy), this is also the case for Shanghai which had the best flood defense measures in China, but experienced several floods in recent years due to overflow and breached dike (Ke, 2014). In a word, it supposes that the (univariate) return period of 10'000 years used in this application should be more frequent (less than 10'000 years when using multivariate return periods) for flood events in reality, unfortunately there is still no such multivariate return period assessment for Shanghai's flood risk. As such, we select an extreme flood depth scenario (a 10'000-year return period flood) to illustrate how many building asset values distributed in the flood extent under this scenario. Moreover, we cannot acquire the inundation depth data for each return period for Shanghai (due to data issues and also the China's circumstances in flood risk assessment).

Minor points 1) "Methodology" is not correct in this context; authors should use "method" instead. "Methodology" pertains to the "study of methods". This issue applies to paper's entire text.

Response: Thanks for your reminding. We will correct this mistake in the revised manuscript according to your suggestion.

2) Authors should pay attention to properly define abbreviations. E.g., "LULC" is never defined; "BFA" is defined only in the Abstract, and should be redefined when first used within the paper (i.e., at page 2, line 20).

Response: Thanks for indicating this problem. We will redefine the abbreviation when first used within the paper in the revised manuscript.
3) Abstract. What does the "immense analytical flexibility" refer to? To the usefulness of the exposure dataset? It seems to me an overstatement.

Response: Yes, what we want to prove is the usefulness of the exposure dataset. We will improve this expression according to your comment in the revised manuscript.

4) Pag. 2, l 2-3: "The main cause for this uncertainty is : : : of water inundation depth" could be supported by additional references, e.g. ": : :of water inundation depth, both in urban and rural areas (e.g., Apel et al., 2016 and Viero et al., 2014, respectively)".

Response: Thanks for your suggestion. We will add these references in the revised manuscript.

5) Fig. 1: a horizontal line is missing in panels b and e, respectively. Please consider increasing the thickness of the grid lines in the lateral panels. Please control "townership".

Response: Thanks for your reminding. We will correct this mistake in the revised manuscript.

6) Sec. 2.2: Point (4) is not devoted to produce the high resolution building asset value map, as it concerns the test for the application of the final result of the proposed method.

Response: Thanks for your reminding. You are right, the proposed method is what we should concentrate on, we will change the description about point (4).

7) The names of the variables that appear in the (few) equations are very little informative, and should be chosen with greater care. In many cases, the names of variables should be exchanged with the subscript. Indeed, the name of the variable commonly identifies the kind of measure (e.g., A for areas, C for costs, and so on: : :), and the subscripts should provide further specifications (e.g., the level at which the area refer to, such as district or township).

Response: Thanks for your suggestion. We will modify the names of the variables that appear in the equations, and explain clearly its exact meaning about the subscripts.

8) Eq. (3): ' is not defined. What does the summation refer to?

Response: Thanks for your reminding. We will supplement the definition in the revised manuscript.

9) Eq. 5: unit, not uint.

Response: Thanks for your reminding. We will correct this mistake in the revised manuscript.

Additional references Apel, H., Martìnez Trepat, O., Hung, N. N., Chinh, D. T., Merz, B., Dung, N. V., 2016. Combined fluvial and pluvial urban flood hazard analysis: concept development and application to Can Tho city, Mekong Delta, Vietnam, Nat. Hazards Earth Syst. Sci., 16, 941–961, doi:10.5194/nhess-16-941-2016. Viero, D. P., Peruzzo, P., Carniello, L., Defina, A., 2016. Integrated mathematical modeling of hydrological and hydrodynamic response to rainfall events in rural lowland catchments, Water Resources Research, 50(7), 5941–5957, doi:10.1002/2013WR014293.

Response: Thanks for providing the related reference information.

Thank you again for your comments. These comments are extremely helpful for improving the manuscript.

Please also note the supplement to this comment:
http://www.nat-hazards-earth-syst-sci-discuss.net/nhess-2017-17/nhess-2017-17-AC2-supplement.pdf

---

## Author Comment (AC3) · 2 May 2017

We thank the referee 2 for his/her careful reading of the paper, and we have found the comments extremely useful while working on a new draft of the paper with substantial changes. We have adopted the recommendations from referee 2.

Response to anonymous Referee #2

The paper presents a method to disaggregate (i.e., downscale) building assets values from census information at the district level to a finer resolution (grid of 2.5 × 2.5 m) using information on population distribution (from LandScan, resolution of ∼ 800m) and building footprint maps. As a case study, the proposed method is applied to the City

of Shanghai (China). The topic is relevant and certainly meets the potential interest of the readers of NHESS. However, I have major concerns with the method proposed and with some of its basic assumptions. Moreover, the presentation of the paper is too lacking to allow a deep and clear understanding of both the scientific significance and the novelty of the work. For these reasons, I suggest rejection and resubmission of a new, improved paper.

The points that need to be considerably improved are detailed in the following, along with some minor comments.

Response: Thanks for your comments, we will have responses for each comment as follows.

Major points 1) The paper need a thorough English revision, which now is not acceptable for publication. Lot of typos, awkward and incorrect sentences are present in the text, and should be fixed preferably with the help of a native speaker.

Response: Thanks for your advice, we will check the English language issues in the revised manuscript, and will be looking for partners whose native language is English to help modify the language problem, or using English language editing service.

2) The use of terminology needs much care. Clear definitions should be given before describing the method. E.g., does the buildings floor area (BFA) from census refer to a single storey or to all the storeys of a building? Moreover, the procedure should be better outlined introducing clear notation and using a suitable number of equations. Importantly, the variables in every equation should be unequivocally linked to the quantities referred to in the text. E.g., I really did not understand what variable, in the equations shown, denotes the BFA.

Response: Thank you for your suggestion. We have adopted your suggestions: a) adding clear definition of our terminology, including the building floor area (BFA) and floor area ratio (FAR): "BFA refers to the total floor area inside the building envelope,

including the external walls, and excluding the roof, floor area ratio (FAR) is the ratio of a building's total floor area (BFA) relative to the size of the piece of land which it is built upon." b) we will improve this in the methodology section, including the modification of the names of the variables which appear in the equations and we will improve the explanation of the subscripts.

3) The description of the overall method is very tangled, and it should be made considerably clearer. Specifically, the description of available data, of their use, and of the overall method are all mixed together, in a way that it is very difficult to grasp a clear understanding of the method proposed by the Authors.

Response: We understand that the current method was slightly unclear. We will disentangle the available data and the method by adding a separate section, i.e., Section 2.2 Data, to explain clearly the description of available data (listed in Table 1) and its use. This will be followed by Section 2.3 Method for this study.

4) The method used to evaluate of the accuracy of the building footprints is dubious. For example, based on the definition of the accuracy ratio (Eq. 1), if the correspondence between vector building footprints and aerial images is equal to 51% in each one of the 25 cells (i.e., a total discrepancy of 49%), the accuracy ratio for the entire place is equal to 1! Moreover, Fig. 1 suggests that the building area from vector building footprint always underestimates the building area provided by the aerial images.

Response: Thanks for your comments on precision validation of building footprints data. On the one hand, we mainly used artificial vision judgement to decide the accuracy ratio of cells, which is a little subjective. However, under no better quantitative method for selection, artificial vision judgement is reasonable to validate the precision of the building footprints data. On the other hand, we agree with your reminder that the building area from vector building footprints always underestimates the building area provided by the aerial images, and we will add this concern to the main text for building footprint precision validation (section XX).

5) I have concerns about the resolution of the final grid (2.5 x 2.5 m) given the resolution of the input data. The buildings footprints are the only data comparable in terms of resolution (as they are provided in vector form), whereas the assumption of invariant distributions at finer scales are incompatible with the final resolution of the grid, as clearly explained and demonstrated by, e.g., Figueiredo and Martina (2016).

Response: We understand your concerns about the data resolution. Firstly, especially for developing countries (such as China), high-resolution land use data is always an obstacle in flood risk modelling. As a way to solve this obstacle, we resort to use high-resolution building footprints data (i.e., rasterized to 2.5 x 2.5 m), which provide exact information of building distribution. However, due to population density grid data limitations, we have to assume invariant distributions within an extent of approximately 800 x 800 m when disaggregating the building asset value to grid cells (using LandScan population density as ancillary data). In other words, the real resolution of the disaggregated building asset value map is not more than 800 x 800 m, we just show the final output in a 2.5 x 2.5m resolution to highlight the building distribution. As Figueiredo and Martina (2016) indicated "lower resolutions of the exposure model will in general lead to an overestimation of the affected buildings" in flood exposure estimation, which is also the intention for this study, i.e., improve the exposure data quality in natural disaster risk estimation. We will add a discussion for this limitation in the Discussion section.

6) Sec. 2.3.3 "Valuation of building assets": the Authors refers to reconstruction costs and to construction costs without a clear distinction of the two. Moreover, flooding of nearly flat urban areas is unlikely to cause the reconstruction of buildings (i.e., walls, etc., which are accounted for in the construction cost). Rather, major economic losses are linked to damage of goods. Moreover, damages are likely to affect only the ground floor, so I don't understand the reason why high-storey buildings should have a loss per unit area that is more than twice that of a low-storey building.

Response: Thanks for pointing out the uncertainty on simulation of flood damage sce-

nario. We agree with your concerns. Firstly, for Sec. 2.3.3 Valuation of building assets, which aims at producing building assets value map for natural disaster risk assessment, such as earthquakes, tropical cyclone, sea level rise, and also floods. So, according to your comments, we consider change the title from "Building Asset Value Mapping in Support of Flood Risk Assessment . . ." into "Building Asset Value Mapping in Support of Natural Disaster Risk Assessment . . .". We will use "reconstruction costs" for the valuation of building assets according to your reminding. Meanwhile, as for the complexity of flood loss modelling, we consider deleted such content as it is not the main focus of this study, and we just want to show how flexible of the disaggregated building asset value map can be used in exposure assessment of floods in the Discussion section, but do not simulate the damages.

Response: We agree that using both construction and reconstruction costs might cause some confusion. In this paper, however, we clearly state the difference between the replacement and reconstruction costs. This is, in our opinion, what is being mixed up the most in literature. In that respect, both 'replacement' and 'reconstruction' will have construction costs (the costs of actually doing the job of reconstruction/replacement of the building).

7) Evaluation of the proposed method. Fig.5 shows a comparison between modeled and real statistical building floor area (BFA). Despite the fact that the comparison is carried out at the district level (i.e., at a very coarse resolution), substantial discrepancies are shown, particularly for small values of BFA. Why?

Response: Thank you for your comment. Firstly, as shown in Figure 5 and Table 2 in the manuscript, for most of the districts the total BFA is less than 50 km2. For each district in each building category (i.e., low-, medium- or high-rise building), we compare the census statistical value and modeled value (shown in Figure 6). As explained in the manuscript, "The figure also shows that most of the district's medium-rise BFA is slightly overestimated, whereas both low-rise and high-rise BFA are slightly underestimated for most of the districts. The spatial resolution of the LandScan population

density data is approximately 800m for Shanghai. This coarse resolution smooths out the high- and low- population density area compared to a 2.5m spatial resolution of the building footprint map that is used here. This may be the main reason for the overestimation of medium-rise BFA in most of the districts of Shanghai."

Moreover, the population density is not perfectly correlated with building floor area, such as villa's in residential areas and 'villages' in the city/. The spatial heterogeneity for this correlation should be even larger at a finer spatial scale. Substantial discrepancies are higher when the spatial resolution is higher (as shown in Figure 6e below, which will be added in the revised manuscript).

Figure 6: Building floor area validation by different grid cell size in part of the downtown area of Shanghai. a) Real building height in the downtown area of Shanghai (i.e., the sampling area for validation). b) Relationship between building height and number of storeys in Shanghai acquired by random building surveys. c) Real building floor area calculated by number of storeys via a) and b) multiplied with occupied area of the building footprint. d) Modeled building floor area as described above in this article. e) Distribution of relative error between modeled BFA and Real BFA by five different grid cell size. f) Relationship between modeled BFA and Real BFA for regular grid cell size of 1600 m. g) Spatial relative error distribution between modeled BFA and Real BFA for regular grid cell size of 1600 m.

8) The flooding scenario used in the final application of the exposure dataset in order to produce estimates of economic losses is far from reality. The return period (10'000 years) is extremely high if compared with common return periods used in the engineering practice. The text says that no flood protection are assumed in the modeling of flooding. It is not clear if flood protections (e.g., levee) actually exist but they have been disregarded in the calculation of the flooding scenario, or if they do not exist (if this is the case, why saying "assuming no flood protection"?).

Response: Thank you for your comment. Firstly, for Shanghai, flood protections (e.g.,

levee) actually exist but they have been disregarded in the calculation of the used flooding scenario (as used in this study). The core reason of using this (external) flood scenario is to show how the use of the disaggregated building asset value map in an exposure assessment to natural disasters. Because the high complexity in flood risk assessment, there are large uncertainties in the flood risk estimation. For flood return period estimation, for instance, much attention has been paid to univariate flood frequency analysis. However, flood risk in Shanghai is a multivariate process which depends on several random variables, such as sea-level rise, high tides, tropical cyclones and floods in the Upper Yangtze River. In the bivariate joint probability distribution for the return period analysis of dust storms (Li et al., Risk Analysis, 33(1), 2013), the results indicated that an 81-year return period dust storm will change to a 4-year return period dust storm (when) using bivariate analysis (Copula-based modelling) compare with using univariate analysis. Shanghai, which had the best flood defense measures in China experienced several floods in recent years due to overflow and dike breaches (Ke, 2014). More specifically, this means that the (univariate) return period of 10'000 years used in this application may be more frequent (less than 10'000 years when using multivariate return periods) for flood events in reality. Unfortunately there is still no such multivariate return period assessment for Shanghai's flood risk. As such, we select an extreme flood depth scenario (a 10'000-year return period flood) to illustrate how much of Shanghai's building asset values is exposed to flooding. Moreover, we cannot acquire the inundation depth data for each return period for Shanghai (due to data issues and also the China's circumstances in flood risk assessment).

Minor points 1) "Methodology" is not correct in this context; authors should use "method" instead. "Methodology" pertains to the "study of methods". This issue applies to paper's entire text.

Response: Thank you for your reminding. We will correct this mistake in the revised manuscript according to your suggestion.

2) Authors should pay attention to properly define abbreviations. E.g., "LULC" is never

defined; "BFA" is defined only in the Abstract, and should be redefined when first used within the paper (i.e., at page 2, line 20).

Response: Thank you for indicating this problem. We will redefine the abbreviation when first used within the paper in the revised manuscript.

3) Abstract. What does the "immense analytical flexibility" refer to? To the usefulness of the exposure dataset? It seems to me an overstatement.

Response: Yes, what we want to prove is the usefulness of the exposure dataset. We will improve this expression according to your comment in the revised manuscript.

4) Pag. 2, l 2-3: "The main cause for this uncertainty is : : : of water inundation depth" could be supported by additional references, e.g. ": : :of water inundation depth, both in urban and rural areas (e.g., Apel et al., 2016 and Viero et al., 2014, respectively)".

Response: Thank you for your suggestion. We will add these references in the revised manuscript.

5) Fig. 1: a horizontal line is missing in panels b and e, respectively. Please consider increasing the thickness of the grid lines in the lateral panels. Please control "township".

Response: Thank you for your reminding. We will correct this mistake in the revised manuscript.

6) Sec. 2.2: Point (4) is not devoted to produce the high resolution building asset value map, as it concerns the test for the application of the final result of the proposed method.

Response: Thank you for your reminding. You are right, the proposed method is what we should concentrate on, we will change the description about point (4).

7) The names of the variables that appear in the (few) equations are very little informative, and should be chosen with greater care. In many cases, the names of variables

should be exchanged with the subscript. Indeed, the name of the variable commonly identifies the kind of measure (e.g., A for areas, C for costs, and so on: : :), and the subscripts should provide further specifications (e.g., the level at which the area refer to, such as district or township).

Response: Thank you for your suggestion. We will modify the names of the variables that appear in the equations, and explain clearly its exact meaning about the subscripts.

8) Eq. (3): ' is not defined. What does the summation refer to?

Response: Thanks for your reminding. We will supplement the definition in the revised manuscript.

9) Eq. 5: unit, not uint.

Response: Thanks for your reminding. We will correct this mistake in the revised manuscript.

Additional references Apel, H., Martìnez Trepat, O., Hung, N. N., Chinh, D. T., Merz, B., Dung, N. V., 2016. Combined fluvial and pluvial urban flood hazard analysis: concept development and application to Can Tho city, Mekong Delta, Vietnam, Nat. Hazards Earth Syst. Sci., 16, 941–961, doi:10.5194/nhess-16-941-2016. Viero, D. P., Peruzzo, P., Carniello, L., Defina, A., 2016. Integrated mathematical modeling of hydrological and hydrodynamic response to rainfall events in rural lowland catchments, Water Resources Research, 50(7), 5941–5957, doi:10.1002/2013WR014293.

Response: Thanks for providing the related reference information.

Thank you again for your comments. These comments are extremely helpful for improving the manuscript.

Please also note the supplement to this comment:
http://www.nat-hazards-earth-syst-sci-discuss.net/nhess-2017-17/nhess-2017-17-AC3-supplement.pdf